# Model-Based Geostatistical Mapping of the Prevalence of *Onchocerca volvulus* in Cameroon between 1971 and 2020

Yannick Niamsi-Emalio[1,2]\*, Hugues C. Nana-Djeunga[1], Claudio Fronterrè[3], Himal Shrestha[4], Georges B. Nko'Ayissi[5], Théophile M. Mpaba Minkat[6], Joseph Kamgno[1,2], María-Gloria Basáñez [7]\*

**1** Higher Institute for Scientific and Medical Research, Yaoundé, Cameroon, **2** Department of Public Health, Faculty of Medicine and Biomedical Sciences, University of Yaoundé I, Yaoundé, Cameroon, **3** Centre for Health Informatics, Computing and Statistics, Lancaster Medical School, Lancaster University, Lancaster, United Kingdom, **4** Department of Microbiology and Immunology, University of Melbourne at the Peter Doherty Institute for Infection and Immunity, Melbourne, Victoria, Australia, **5** Neglected Tropical Diseases National Coordination Unit, Ministry of Public Health, Yaoundé, Cameroon, **6** National Onchocerciasis Control Programme, Ministry of Public Health, Yaoundé, Cameroon, **7** MRC Centre for Global Infectious Disease Analysis and London Centre for Neglected Tropical Disease Research, School of Public Health, Imperial College London, London, United Kingdom

\* emalio2002@yahoo.fr (YNE); m.basanez@imperial.ac.uk (MGB)

## Abstract

### Background

After the closure of the African Programme for Onchocerciasis Control (APOC) in 2015, the Ministry of Public Health of Cameroon has continued implementing annual community-directed treatment with ivermectin (CDTI) in endemic areas. The World Health Organization has proposed that 12 countries be verified for elimination (interruption) of transmission by 2030. Using Rapid Epidemiological Mapping of Onchocerciasis, a baseline geostatistical map of nodule (onchocercoma) prevalence had been generated for APOC countries, indicating high initial endemicity in most regions of Cameroon. After more than two decades of CDTI, infection prevalence remains high in some areas. This study aimed at mapping the spatio-temporal evolution of *Onchocerca volvulus* prevalence from 1971 to 2020 to: i) identify such areas; ii) indicate where alternative and complementary interventions are most needed to accelerate elimination, and iii) improve the projections of transmission models.

### Methodology

A total of 1,404 georeferenced (village-level) prevalence surveys were obtained from published articles; the Expanded Special Project for Elimination of Neglected Tropical Diseases portal for Cameroon; independent researchers and grey literature. These data were used together with bioclimatic layers to generate model-based geostatistical (MBG) maps of microfilarial prevalence for 1971–2000; 2001–2010 and 2011–2020.

provided the original author and source are credited.

**Data availability statement:** All relevant data are within the paper and its Supporting Information files.

**Funding:** M.G.B. acknowledges funding from the MRC Centre for Global Infectious Disease Analysis (MR/X020258/1), funded by the UK Medical Research Council (MRC). This UK-funded award is carried out in the frame of the Global Health EDCTP3 Joint Undertaking. M.G.B. also acknowledges funding by the Bill & Melinda Gates Foundation through the NTD Modelling Consortium (INV-030046). J.K. and M.G.B. are in receipt of a Royal Society International Exchanges Award (IES\R2\181017) for a collaborative project on "Modelling the elimination of onchocerciasis in Cameroon". The funders had no role in study design, data collection and analysis, decision to publish, or preparation of the manuscript.

**Competing interests:** The authors have declared that no competing interests exist.

## Principal findings

Time-period was negatively and statistically significantly associated with prevalence. In 1971–2000 and 2001–2010, prevalence levels were high in most regions and ≥60% in some areas. Mean predicted prevalence declined in 2011–2020, reaching <20% in most areas, but data for this period were sparse, leading to substantial uncertainty. Hotspots were identified in South West, Littoral and Centre regions.

## Conclusions/Significance

Our results are broadly consistent with recent MBG studies and can be used to intensify onchocerciasis control and elimination efforts in areas with persisting transmission, providing spatio-temporal prevalence trends to which transmission models can be fitted to improve projections of onchocerciasis elimination by 2030 and beyond.

## Author summary

Onchocerciasis ('river blindness') is a severely debilitating neglected tropical disease proposed for elimination of transmission by the World Health Organization. The main intervention strategy consists of prolonged treatment of endemic communities with ivermectin. Ivermectin treatment started in Cameroon approximately in the year 2000, but after more than 20 years of treatment the disease remains endemic in many areas of the country. Understanding changes in onchocerciasis prevalence geographically and with time under intervention is crucial for planning where control efforts should be intensified and with which strategies. Geostatistical mapping is an ideal tool for this purpose. Therefore, we produced onchocerciasis predicted prevalence maps for 1971–2000 (before treatment began), 2001–2010 and 2011–2020 using published, unpublished and publicly-available data. Prior to the start of treatment and in the decade following its commencement, the country exhibited high prevalence (more than 60% in many areas). In 2011–2020 predicted prevalence declined in most but not all areas, remaining above 20% in some areas of the South West, Littoral and Centre regions. These hotspots would benefit from more efficacious medications and complementary control measures. Our results can be combined with transmission models to generate infection trends under simulated strategies until 2030 and beyond.

## Introduction

Onchocerciasis is a neglected tropical disease (NTD) proposed for elimination of transmission (EOT) by the World Health Organization (WHO) in its second, 2021–2030 Roadmap on NTDs [1]. The disease is caused by infection with the filarial nematode *Onchocerca volvulus* and is transmitted via the bites of *Simulium* blackflies [2]. Clinical manifestations include skin and eye disease, ultimately leading to blindness and excess mortality of the blind [3,4]. Onchocerciasis is also associated with neuro-hormonal involvement, including Nakalanga syndrome, nodding syndrome and more generally epilepsy [5]. After accounting for blindness, infection (microfilarial) load is significantly and positively associated with relative risk of mortality in a dose-response manner [6], which is statistically significantly greater for children than for adults [7]. The blackfly vectors breed in fast-flowing water bodies such as rivers, rapids and

waterfalls, lending the disease the name of 'river blindness'. The term 'river epilepsy' has also been proposed [8]. Populations living near these water bodies may be exposed to high blackfly biting rates, and sometimes feel obliged to abandon their communities due to the combined effect of blackfly nuisance and disease severity [2]. The disease and socio-economic burden of onchocerciasis have resulted in its consideration as a serious public health problem [9].

To tackle onchocerciasis as a public health problem, regional control programmes were implemented particularly in sub-Saharan Africa (SSA), where more than 99% of the cases occur [2]. The first of such programmes was the Onchocerciasis Control Programme in West Africa (OCP, 1974–2002), which started as a large-scale vector control programme using weekly aerial larviciding of blackfly breeding sites [10,11]. In 1987, the endectocide drug ivermectin was approved for use in humans and donated (as Mectizan) for as long as necessary by Merck & Co., Inc. [12]. Mass administration of ivermectin started as a complement to vector control or as the only intervention (in some areas of the Western Extension) in the late 1980s, initially delivered by mobile teams, and since the late 1990s via community-directed treatment with ivermectin (CDTI) [10]. The OCP covered a total of 11 countries in West Africa (excluding Liberia) and aimed at eliminating blindness as a public health problem [10,11]. The African Programme for Onchocerciasis Control (APOC, 1995–2015) covered the remaining 20 endemic countries in Africa, with annual CDTI as the main intervention [13,14].

Ivermectin impacts on infection, morbidity and transmission because it exerts a microfilaricidal and embryostatic effect on *O. volvulus* [15], also potentially having a cumulative sterilizing and/or macrofilaricidal effect upon repeated exposure of adult worms to the drug [16,17]. Although APOC started as a control programme aiming at onchocerciasis elimination as a public health problem (EPHP), in 2010 it shifted its goal from EPHP to EOT [18], encouraged by the results obtained with (annual or biannual) mass drug administration (MDA) of ivermectin alone in some foci of the Western Extension of the former OCP, namely the River Bakoye, River Falémé and River Gambia foci in Mali and Senegal, where 15–17 years of MDA were deemed to have led to EOT [19,20]. To achieve country-wide EOT, all endemic areas would need to be under MDA for long enough to interrupt transmission, including low-prevalence (hypoendemic) areas. Originally, APOC had prioritized meso- and hyperendemic foci (i.e., those with moderate to high prevalence; see below for definitions of endemicity). Upon the closure of APOC in 2015, onchocerciasis control and elimination activities were devolved to the ministries of health of the endemic countries with technical support by the Expanded Special Project for Elimination of Neglected Tropical Diseases (ESPEN) [21].

In Cameroon, the Ministry of Public Health (MINSANTE), through the national onchocerciasis control programme (NOCP) became responsible for (annual) CDTI. Programmatic treatment indicators, such as geographical and therapeutic coverage, are calculated by officials in health districts before being sent to the regional and national authorities. In addition to onchocerciasis, loiasis is endemic in Cameroon, and its distribution covers almost the entire lower half of the country [22,23]. Loiasis is another filarial infection caused by *Loa loa* and transmitted through the bites of *Chrysops* tabanid flies. Studies have documented the occurrence of severe adverse events (SAEs) in individuals treated with ivermectin for onchocerciasis who harboured high *L. loa* microfilarial loads [24,25], leading to fatal encephalopathies or irreversible neurological sequelae. Therefore, risk-benefit considerations have precluded the implementation of standard CDTI in areas hypoendemic for onchocerciasis which are co-endemic with highly-endemic loiasis, but have recommended that CDTI be implemented (with strengthened pharmacovigilance) in meso- and hyperendemic areas [26].

In onchocerciasis, endemicity is categorized according to all-ages microfilarial prevalence into: i) hypoendemicity (>0% – <35%), ii) mesoendemicity (≥35% – <60%) and iii) hyperendemicity (≥60%) owing to their relationship with blindness prevalence [27]; a holoendemicity

category (microfilarial prevalence ≥80%) has also been used [28,29]. Other categorizations, based on onchocercal nodule prevalence in adult males for Rapid Epidemiological Mapping of Onchocerciasis (REMO) refer to hypoendemicity as >0% – 20% nodule prevalence (approximately equivalent to >0% – <35% microfilarial prevalence in those aged ≥5 years), mesoendemicity as >20% – <40% nodule prevalence (microfilarial prevalence ≥35% – <60%), and hyperendemicity as ≥40% nodule prevalence (≥60% microfilarial prevalence) [30].

Transmission dynamics modelling of onchocerciasis has shown that the feasibility of and programme duration required to achieve EOT are strongly determined by baseline (pre-control) endemicity level, history of control interventions, and CDTI coverage and adherence among others [31]. A recent systematic review and meta-analysis of published literature reporting on onchocerciasis status across SSA has largely confirmed these modelling insights [29]. The higher the pre-intervention endemicity level (reflective of high vector biting rates), and the lower the treatment coverage and adherence, the more difficult it becomes to reach EOT. Therefore, understanding the geographical distribution and level of infection endemicity becomes paramount for the planning and monitoring of control interventions.

In Cameroon, a baseline REMO map was initially presented [32] and later refined [33], with the latter using model-based geostatistics (MBG) to analyze the REMO data and generate high-resolution maps of the predicted nodule prevalence and of the probability that the true pre-control nodule prevalence exceeded the threshold of 20% (i.e., that onchocerciasis was at least mesoendemic prior to the commencement of control). More recently, Bayesian MBG has been applied to produce spatially continuous estimates of microfilarial prevalence (all ages) for 2000–2018 [34]. This study reported that, for Cameroon, mean estimates exceeded 5% prevalence at the national level and 25% in some focal areas, the latter including those ineligible for standard CDTI due to loiasis co-endemicity [34]. However, in eligible areas other factors, such as low treatment adherence, play a fundamental role in maintaining high prevalence despite prolonged CDTI. Studies have reported an average proportion of individuals not taking treatment in the last 5 years of a 15-year programme as high as approximately 20% in the Centre and Littoral regions [35] and 30% in the West region [36].

In this paper we present a geostatistical model fitted to *O. volvulus* microfilarial prevalence across space and time in Cameroon to visualize the evolution of onchocerciasis control efforts carried out by the NOCP. The ultimate aim will be to combine MBG and transmission dynamics modelling to identify areas of the country that may reach EOT with the current CDTI approach and those that will necessitate alternative and/or complementary strategies towards the 2030 time-horizon and beyond.

## Methods

### Data

Data for the period 1971–2020 were obtained following a systematic review of published studies on onchocerciasis endemicity in Cameroon (with protocol presented by Nana-Djeunga et al. [37]), complemented with unpublished survey data collected by co-authors (HCN-D, JK); the NOCP (GBN, TMMM), and data from the ESPEN portal for Cameroon [38]. The results of this review will be presented in a separate publication. To identify data duplicates, records were compared according to region, health district, village, period, and number of people surveyed. After duplicates were removed, the data comprised 983 (70%) nodule prevalence surveys and 421 (30%) microfilarial prevalence surveys, for a total of 1,404 (village-level) georeferenced surveys during the overall (1971–2020) study interval. Of the 983 nodule prevalence surveys, 842 (86%) were obtained from the ESPEN portal, and 141 (14%) from published literature. Of the 421 microfilarial prevalence surveys, 122 (29%) were from

ESPEN, 204 (48%) from the literature, and 95 (23%) from unpublished surveys. Microfilarial prevalence surveys were conducted by the skin-snip microscopy method. For this, two skin snips (one from each iliac crest) were taken from each examined individual (aged ≥5 years) with a 2-mm Holth corneoscleral punch and incubated in saline solution to allow emergence of the microfilariae, whose presence was detected by examining the incubation medium under a microscope [37]. Nodule prevalence surveys were conducted by detecting the presence of palpable (subcutaneous) onchocercomas in adult males residing in the villages [30,32,33].

Environmental data, including altitude (elevation), topography, normalized difference vegetation index (NDVI), and bioclimatic variables (BIO1 to BIO19) were downloaded from the National Aeronautics and Space Administration (NASA) portal [39] at 5 km x 5 km spatial resolution. The definition of the bioclimatic variables can be found in Table A in S1 File.

## Statistical analysis

### Conversion of nodule into microfilarial prevalence and its stratification, and data mapping

As the data collated included both microfilarial and nodule prevalence, nodule prevalence (in samples of adult males aged ≥20 years [30,32,33]) was converted into microfilarial prevalence (in those aged ≥5 years) [40] (Text A in S1 File).

The maps for Cameroon were downloaded from https://gadm.org/download_country_v3.html version 3.6, under the https://gadm.org/license.html licence, which allows for the free use of data for academic purposes including creating maps for research articles published in Open Access journals. The data were plotted onto these maps using the ggspatial (https://cran.r-project.org/web/packages/ggspatial/index.html) and sf (https://cran.r-project.org/web/packages/sf/index.html) packages of the R software version 4.4.2, for the 10 regions of Cameroon, namely, Far North, North and Adamaoua in the upper half of the country, and North-West, South-West, West, Littoral, Centre, South and East in the lower half, from west to east. Different maps were prepared for i) pre-intervention (baseline) data: 1971–2000; ii) 2001–2005; iii) 2006–2010; iv) 2011–2015; and v) 2016–2020. For the pre-control map, the microfilarial prevalence endemicity levels described above [27] were used for stratification. However, and to better discern locations with low prevalence in both the pre-control and subsequent maps, the additional category of 0% was used for those records reporting 0% prevalence, and the >0 – <35% category was subdivided into >0% – <10% and ≥10% – <35%. We identified a total of 571 georeferenced prevalence surveys for the 1971–2000 period; 410 for 2001–2005; 98 for 2006–2010; 261 for 2011–2015 and 64 for 2016–2020, for a total of 1,404.

### Spatial unit and time periods

We used village-level surveys (our spatial unit). The years of data collation were grouped according to whether the village-level data had been collected prior to CDTI (pre-control) or following its implementation. For the pre-control period, data were grouped into a single (1971–2000) period. Following CDTI implementation, four 5-year intervals were used for descriptive analyses: 2001–2005, 2006–2010, 2011–2015 and 2016–2020. However, since prevalence surveys for 2006–2010 and 2016–2020 were scarce (less than 100), only the baseline (1971–2000) period and two 10-year classes (2001–2010 and 2011–2020) were used for the formal MBG analysis.

### Geostatistical model

In order to model the spatial variation of microfilarial prevalence, a MBG approach was used to take into account the spatial correlation between observations and the effects of

environmental covariates [41,42]. We followed procedures previously described [28,41]. Firstly, we used logistic regression—a type of generalized linear model (GLM) for binary dependent variables—with a stepwise method based on minimising the Akaike Information Criterion (AIC), to identify statistically significant environmental covariates, setting the significance level at 0.05. Multicollinearity checks were carried out using analysis of variance inflation factors (VIF) [43]. Predictors with VIF greater than 10 were removed (Text B in S1 File). Secondly, from the residuals of these binomial GLM models, semi-variograms were prepared using the Matérn correlation function. Finally, the covariates retained were included in the spatial binomial logistic regression model, implemented using the binomial.logistic. MCML function of the PrevMAP R package version 1.5.4 [44], which estimates model parameters using Monte Carlo maximum likelihood (MCML) [45,46]. This process was repeated until convergence was reached and for all different time periods. In the first instance, the model was fitted to the overall, 1971–2020 interval to assess the effect of intervention period. Subsequently, the model was fitted separately to the baseline (1971–2000), 2001–2010 and 2011–2020 periods. All analyses were performed with R software version 4.4.2 [47]. Text B in S1 File provides full details of the MBG approach and parameter estimation procedures as well as the rationale for spatial prediction, and—of particular interest for detection of hotspots— the calculation of exceedance probability, which was used to prepare predictive maps that microfilarial prevalence is at least 35%. The resolution of the output maps is 5 km x 5km.

## Results

### Distribution of (survey) microfilarial prevalence

Fig 1 shows the distribution of surveyed microfilarial prevalence at baseline (1971–2000).

Fig 1 shows that there were hyperendemic villages in all regions of the country, particularly in the North, West, South-West and Centre regions.

Fig 2 presents the distribution of (survey) prevalence for the 2002–2005, 2006–2010, 2011–2015 and 2016–2020 periods. As mentioned above, prevalence surveys were sparse for the 2006–2010 and 2016–2020 periods. During the 2001–2005 period (Fig 2A), microfilarial prevalence was not dissimilar to that of the pre-intervention period. For the 2006–2010 period (Fig 2B), some prevalence surveys were identified for the North, West, Littoral and Centre regions, whilst no data were located for the Far North, Adamaoua, East, South, North-West and South-West regions. For the 2011–2015 period (Fig 2C), after 10–15 years of CDTI, high microfilarial prevalence was still recorded in the western part of the country. We were unable to locate survey data for the Far North region. For the 2016–2020 period (Fig 2D), data were identified only in the Adamaoua, Centre and Littoral regions, with the latter indicating high (≥60%) microfilarial prevalence in some surveyed villages.

## Geostatistical analysis

### Variograms

Fig 3 presents the semi-variograms for the spatial autocorrelation in the data of the three periods used for MBG. Over the pre-intervention 1971–2000 period there is evidence of spatial correlation over distances of up to 200 km (Fig 3A), whereas this distance is somewhat shorter for the 2001–2010 period (Fig 3B) and considerably longer for 2011–2020 (Fig 3C).

### Covariates and spatial covariance

Table 1 presents the results of the binomial geostatistical model for microfilarial prevalence for the overall, 1971–2020 interval. The only statistically significant variables

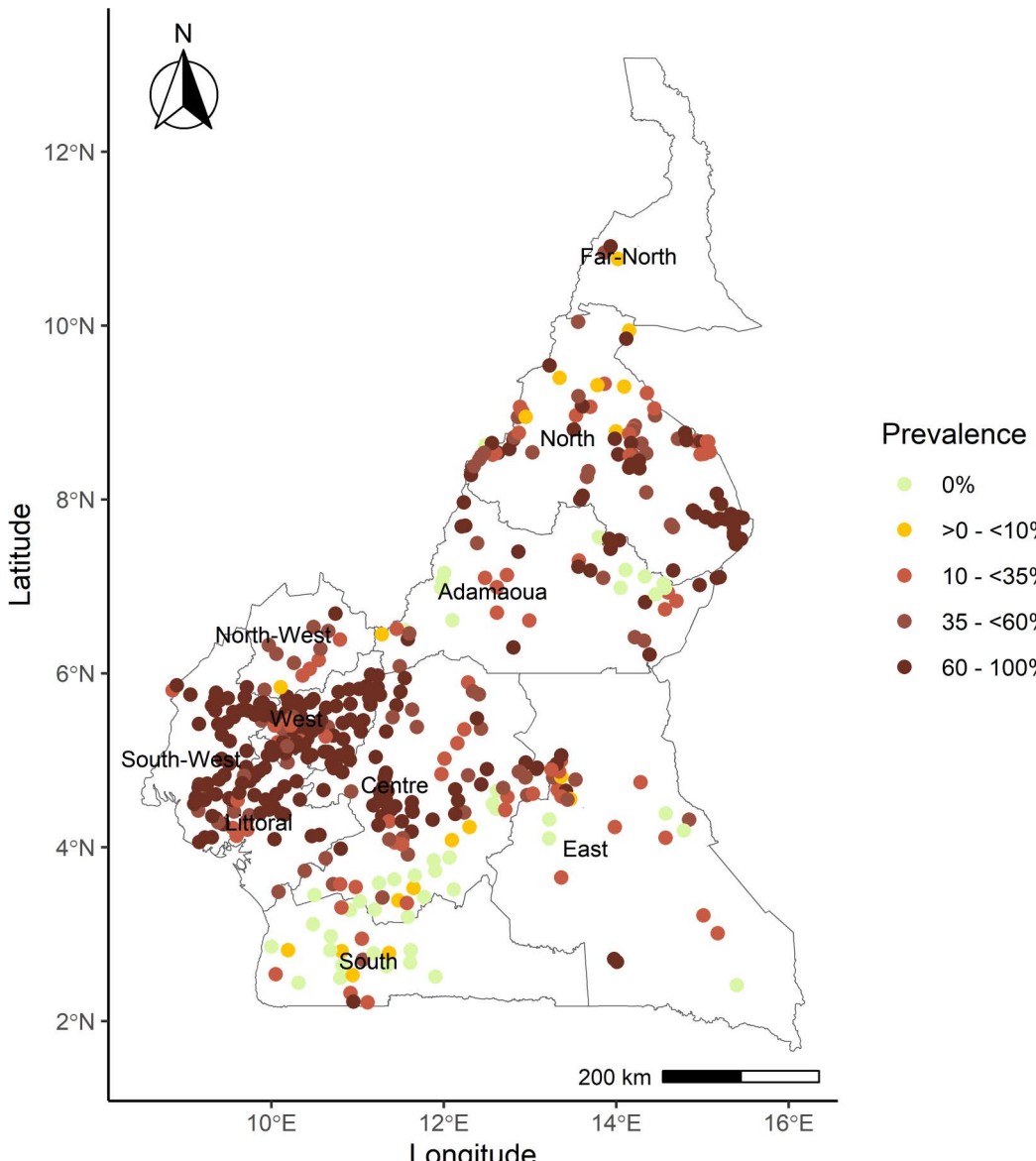

**Fig 1. Map of Cameroon indicating the location of 571 georeferenced locations for which *Onchocerca volvulus* prevalence surveys were identified for the pre-control (1971–2000) period.** Nodule prevalence was converted into microfilarial prevalence [40] where only the former was available (Text A in S1 File), for 388 (68%) surveys. The colour of the circles represents baseline endemicity: from lime-green (0% microfilarial prevalence) to dark brown (hyperendemicity, 60–100% microfilarial prevalence). The hypoendemicity level (>0% – <35% microfilarial prevalence) has been subdivided into >0% – <10% and ≥10% – <35% microfilarial prevalence to facilitate comparison with Fig 2. Base layer of the map available from: https://gadm.org/download_country_v3.html.

were Period and NDVI of the third ecological quarter (June–August). The coefficients for the 2001–2005, 2006–2010, 2011–2015 and 2016–2020 periods were all negative (-0.974, -2.729, -2.697 and -4.455, respectively), and their 95% confidence intervals did not include zero, indicating that compared to the reference, baseline (1971–2000) period, *O. volvulus* microfilarial prevalence has statistically significantly decreased over time in Cameroon. NDVI of the third ecological quarter was statistically significant and

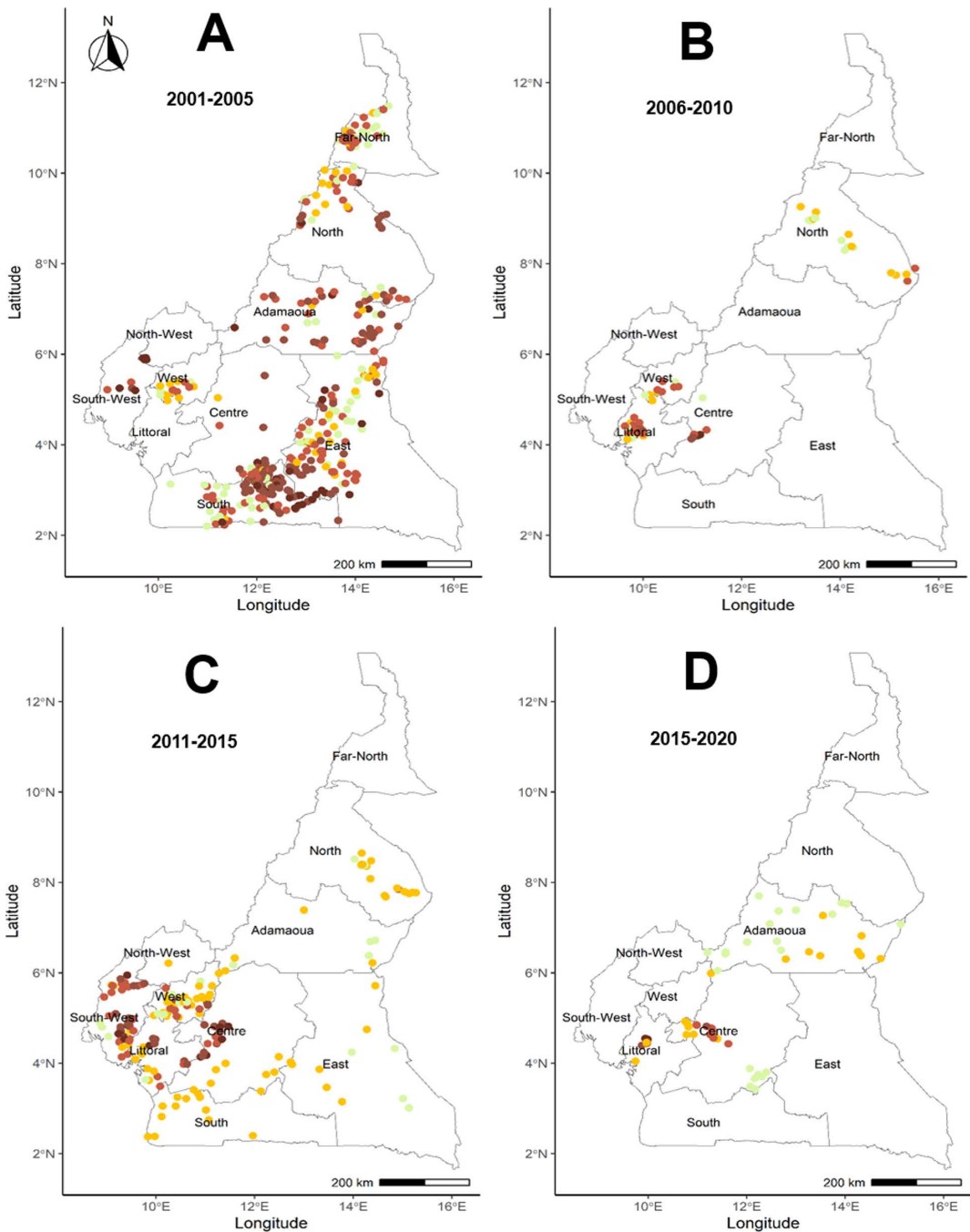

**Fig 2. Map of Cameroon indicating the location of georeferenced *Onchocerca volvulus* prevalence surveys identified for four periods after the commencement of community-directed treatment with ivermectin (CDTI).** Nodule prevalence was converted into microfilarial prevalence [40] where only the former was available (Text A in S1 File), for 595 (71%) surveys. A) 2001–2005 (410 prevalence surveys); B) 2006–2010 (98); C) 2011–2015 (261); D) 2016–2020 (64). Lime-green circles denote surveys reporting 0% prevalence. The >0% – <35% category is subdivided into >0% – <10% and ≥10% – <35%. Base layer of the maps available from: https://gadm.org/download_country_v3.html.

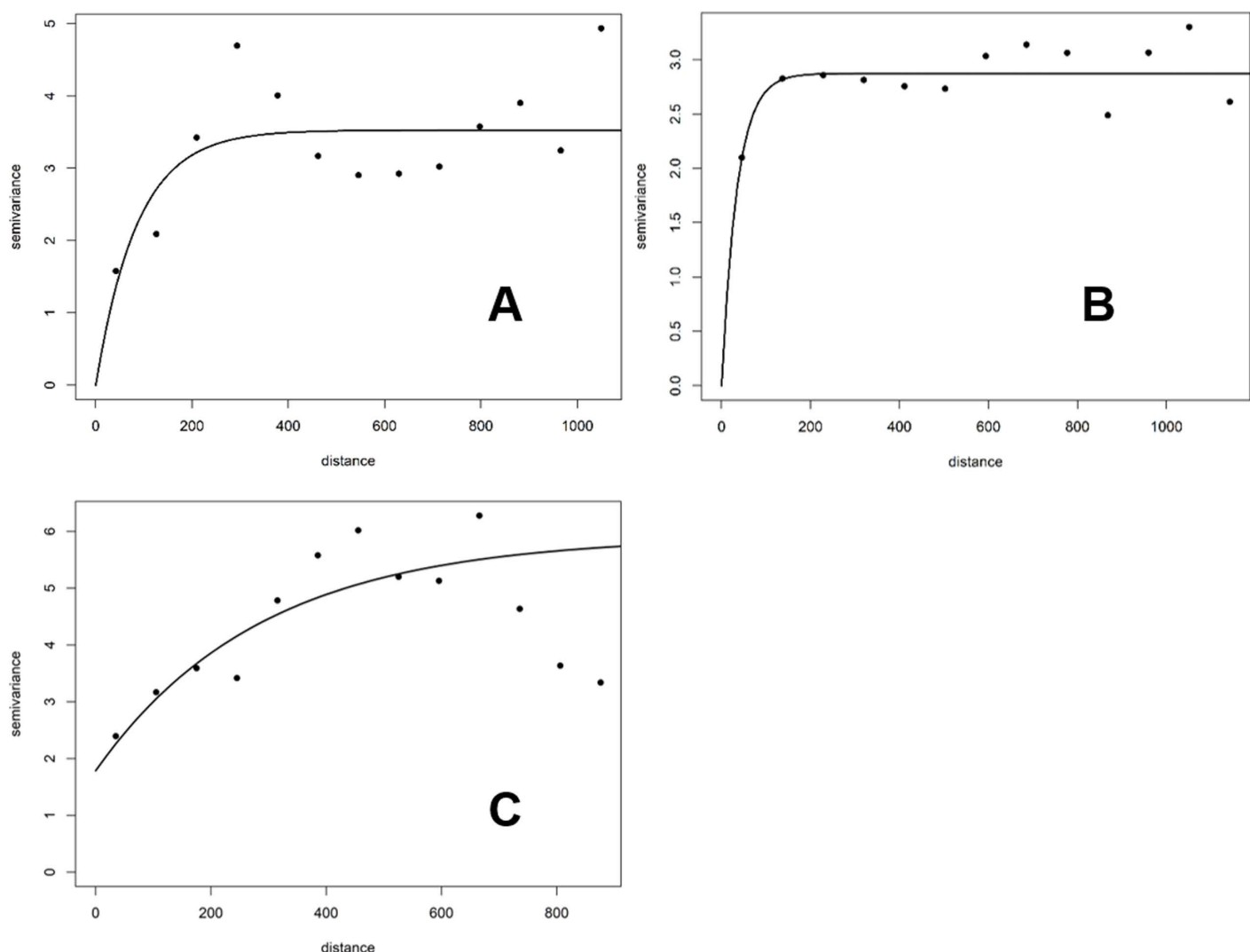

**Fig 3. Empirical semi-variograms for the three periods used in the Model-based geostatistical (MBG) analysis.** Black dots correspond to the empirical logit transformation of the observed prevalence; solid lines correspond to the theoretical semi-variogram. Distances are represented on the horizontal axis (in km); the semivariance on the vertical axis. A) Baseline, 1971–2000, relative nugget = 0; B) 2001–2010, relative nugget = 0; C) 2011–2020, relative nugget = 1.79. These variograms were computed using the residuals from the binomial generalised linear models (GLM) that included the selected covariates for each period (see Methods), with Tables B, C and D in S1 File, respectively, indicating the covariates that were included in the geostatistical models.

negatively associated with microfilarial prevalence (coefficient -2.7, p = 0.0390). The range parameter (67.45 km) and sill (1.92) values confirmed the presence of spatial auto-correlation in the data.

Tables B, C and D in S1 File present the results of the binomial geostatistical models for the baseline (1971–2000), 2001–2010 and 2011–2020 periods, respectively. For the baseline period, BIO3 (isothermality) was negatively and statistically significantly associated with microfilarial prevalence (coefficient = -0.129, p = 0.0218), whereas BIO9 (mean temperature of the driest quarter) was positively and statistically significantly associated (coefficient = 0.181, p = 0.0396). The range parameter was 98.9 km and the sill equal to 2.4 (Table B in S1 File). For the 2001–2010 period, none of the environmental covariates tested showed a statistically significant association with microfilarial prevalence; the range and sill parameters

**Table 1. Parameter estimates and 95% confidence limits (CL) for the binomial geostatistical model of the association between microfilarial prevalence, period and environmental covariates, and spatial covariance parameters over the entire 1971–2020 study period.**

| Variable | Coefficient | Lower 95% CL | Upper 95% CL | p-value |
|---|---|---|---|---|
| Intercept | 2.649 | -6.679 | 11.977 | 0.57781 |
| Period 1971–2000 | Reference | – | – | – |
| 2001–2005 | -0.974 | -1.262 | -0.686 | < 0001 |
| 2006–2010 | -2.729 | -3.172 | -2.286 | < 0001 |
| 2011–2015 | -2.697 | -2.956 | -2.438 | < 0001 |
| 2016–2020 | -4.455 | -4.910 | -4.000 | < 0001 |
| NDVI[1] (3rd quarter Jun–Aug) | -2.708 | -5.278 | -0.138 | 0.0390 |
| NDVI[1] (4th quarter Sep–Nov) | 1.970 | -1.115 | 5.055 | 0.2107 |
| BIO3: Isothermality | -0.040 | -0.136 | 0.056 | 0.4069 |
| BIO4: Temperature seasonality | -0.005 | -0.021 | 0.011 | 0.5023 |
| BIO8: Mean temperature of wettest quarter | 0.029 | -0.110 | 0.168 | 0.6841 |
| BIO14: Precipitation of driest month | -0.003 | -0.042 | 0.036 | 0.8967 |
| BIO16: Precipitation of wettest quarter | 0.000 | -0.002 | 0.002 | 0.9430 |
| BIO18: Precipitation of warmest quarter | 0.001 | -0.003 | 0.005 | 0.4768 |
| BIO19: Precipitation of coldest quarter | 0.000 | -0.002 | 0.002 | 0.8173 |
| $\phi$, range parameter (km) | 67.452 | 67.444 | 67.460 | – |
| $\alpha^2$, Sill | 1.916 | 1.708 | 2.124 | – |

[1]NDVI: Normalised difference vegetation index.

were equal to 43.7 km and 1.9, respectively (Table C in S1 File). For the 2011–2020 period, BIO6 (minimum temperature of the coldest month) was positively and statistically significantly (albeit marginally) associated with microfilarial prevalence (coefficient = 0.443, p = 0.0414), whereas BIO13 (precipitation of the wettest month) was negatively and statistically significantly associated (coefficient = -0.011, p = 0.0157). As for the other periods, there was evidence of spatial autocorrelation, with range and sill parameters equal to, respectively, 56.0 km and 2.4 (Table D in S1 File).

The predicted distribution of (categorised) microfilarial prevalence and accompanying standard error (SE) for the MBG-modelled periods are shown in Fig 4. Fig 4A presents the mean predicted microfilarial prevalence for the 1971–2000 period. It can be seen that, prior to the start of CDTI, predicted prevalence reached values ≥80% in areas of the North, South-West, West, Littoral and Centre regions of the country. The SE values around the predicted prevalence (Fig 4B) are high for the Far North region, where survey data were sparser (likely because the northern and eastern part of this region are swampy areas considered unsuitable for onchocerciasis [32]).

Fig 4C presents the mean predicted microfilarial prevalence for the 2001–2010 period, indicating high prevalence in the Adamaoua, North-West, South-West, Littoral and Centre regions, as well as in the East and South regions. The SE of the predicted prevalence (Fig 4D) is, again, greatest for the Far North region, but also for the North and East regions, the latter without any identified surveys for 2006–2010, as shown in Fig 2B. Fig 4E suggests that predicted microfilarial prevalence in 2011–2020 is much reduced, with pockets of high prevalence remaining in the South-West, Littoral and the western part of the Centre region, but surveys for the 2016–2020 period were very scarce or highly localised, with the exception of the Adamaoua region, for which there was a good spread of surveys indicating low

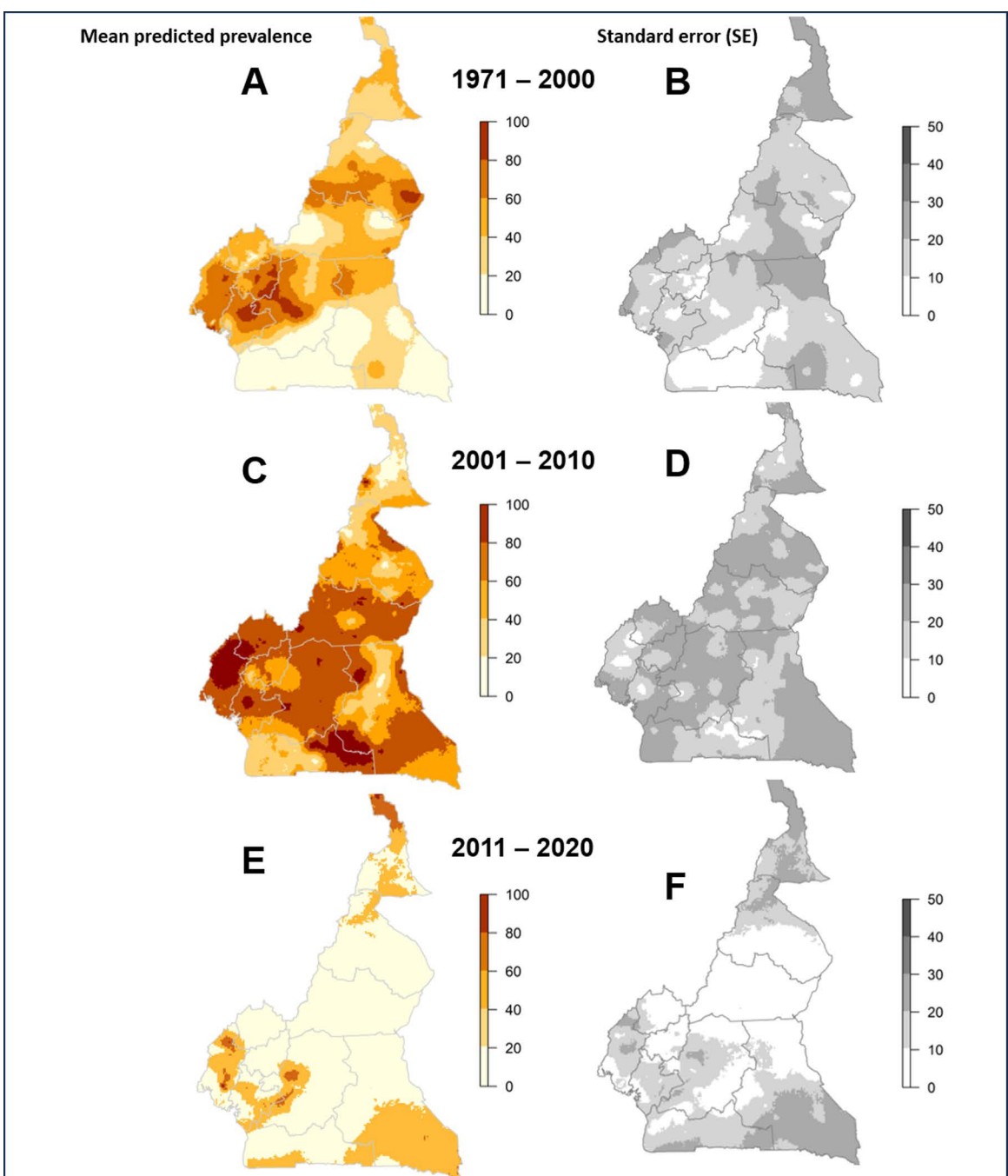

**Fig 4. Distribution of *Onchocerca volvulus* mean predicted microfilarial prevalence and its associated standard error (SE) in Cameroon from the period-specific binomial geostatistical models.** A) and B) are, respectively, predicted microfilarial prevalence and its SE for 1971–2000; C) and D) correspond to 2001–2010; E) and F) correspond to 2011–2020. Base layer of the maps available from: https://gadm.org/download_country_v3.html.

or 0% mean predicted prevalence (Fig 2D). This is reflected in the much lower SE values for Adamaoua (Fig 4F). Higher levels of uncertainty remain for the Far North and the southern part of East region.

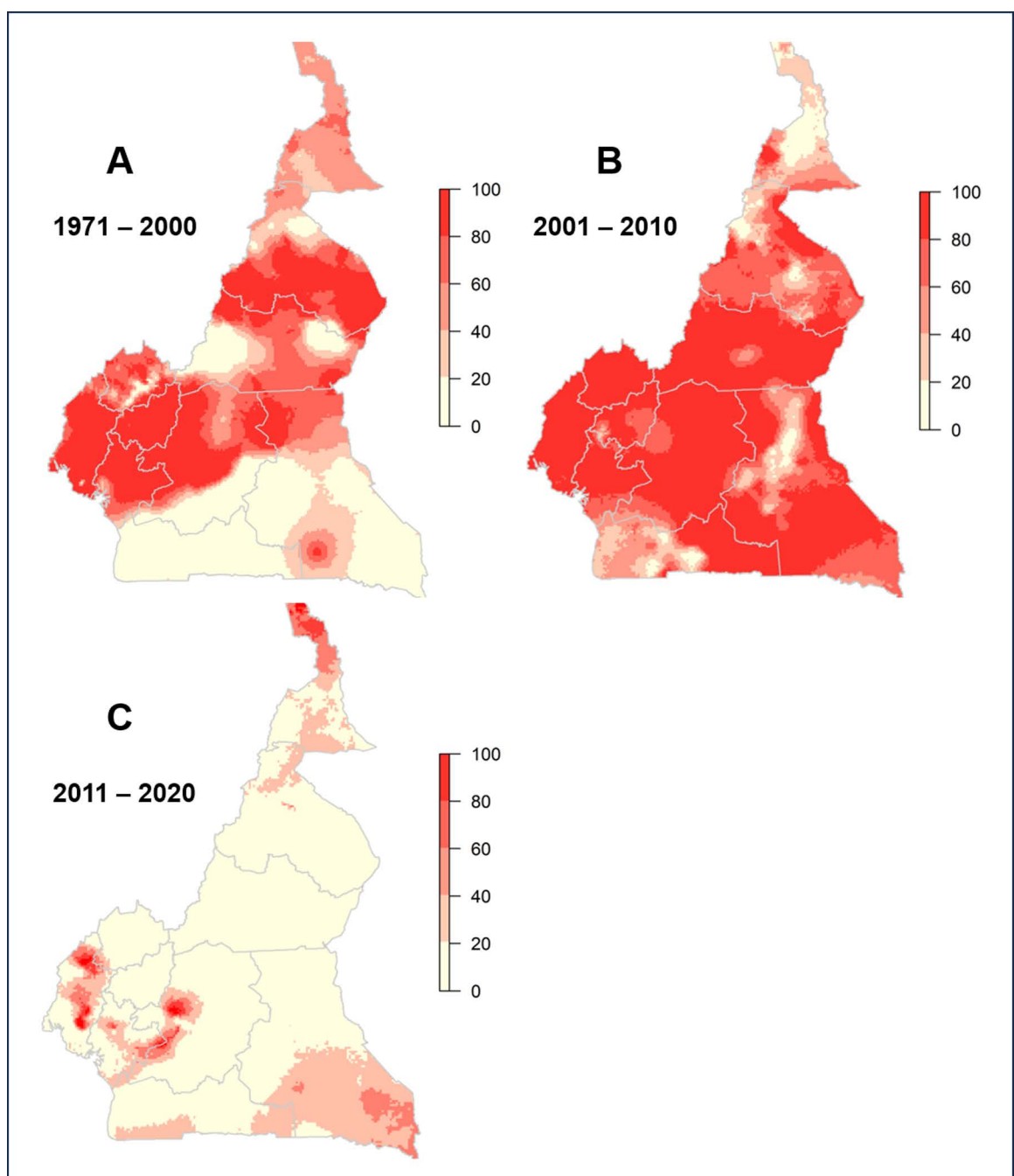

**Fig 5. Predictive inference for threshold exceedance microfilarial prevalence.** The maps show the predictive probability that *Onchocerca volvulus* microfilarial prevalence exceeds ≥35% (i.e., onchocerciasis is at least mesoendemic). A) 1971–2000; B) 2001–2010; C) 2011–2020. Base layer of the maps available from: https://gadm.org/download_country_v3.html.

## Exceedance probability

Fig 5 presents maps of the predictive probability that microfilarial prevalence exceeds the threshold for at least mesoendemicity, of ≥35%. For the 1971–2000 and 2001–2010 periods, the exceedance probability was generally high across the country (Fig 5A and 5B). For

2011–2020 the exceedance probability of being at least mesoendemc remains high in some areas of the South-West, Littoral and Centre regions.

## Discussion

### MBG mapping of onchocerciasis in Cameroon

We have used MBG to generate maps of onchocerciasis predicted prevalence in Cameroon between 1971 and 2020. Since Cameroon was part of APOC, a previous MBG study had mapped the pre-control prevalence of onchocerciasis, as part of a larger exercise to delineate the geographical distribution of *O. volvulus* in all 20 APOC countries prior to the commencement of interventions [33]. However, this study only used nodule palpation data and did not include environmental variables. Earlier mapping studies had been carried out to illustrate the use of REMO with Cameroon as a case study [32]. Recently, MBG has been used to produce continuous maps of microfilarial prevalence across SSA from 2000 to 2018, tracking changes in prevalence through time since the inception of CDTI [34]. In addition, detailed country-specific mapping of the environmental suitability index for onchocerciasis in Africa has been presented [48]. These studies allow us to compare our results for Cameroon.

We identified a total of 1,404 georeferenced prevalence surveys during the overall 1971–2020 study interval (compared to 1,216 for Cameroon in Schmidt et al. [34]). When jointly analysing all the data for 1971–2020, the coefficients for 2001–2005, 2006–2010, 2011–2015 and 2016–2020 were statistically significant and negatively associated with microfilarial prevalence, indicating a decrease in infection prevalence compared to the reference, 1971–2000, pre-intervention period (Table 1). Although our study is a prediction rather than a causal inference study, Text C in S1 File offers a cautious interpretation of the environmental coefficients. Regarding the baseline data, we identified 388 (47.5%) of the 817 nodule surveys [30] mapped in Fig 1 of Zouré et al. [33] in addition to 183 non-REMO (non-nodule prevalence) surveys. The initial REMO study [32] had been based on 349 nodule surveys. Our pre-control mean predicted microfilarial prevalence map (1971–2000, Fig 4A) highlighted areas of high onchocerciasis prevalence in the North, South-West, West, Littoral and Centre regions, in agreement with others [33,34]. We also captured an area of high prevalence in the west of Far North and in the South region (2001–2010, Fig 4C). These are areas where onchocerciasis occurrence (by any parasitological, clinical, serological or other diagnostic) had been located [48]. Our map for the 2001–2010 period indicates areas of high predicted prevalence in the South-West, West, Littoral, Centre, East and South regions, comparable to the MBG map for 2005 presented in Schmidt et al. [34]. For the 2011–2020 period, both the 2010 and 2018 maps of Schmidt et al. [34] indicate a substantial reduction in microfilarial prevalence, yet highlighting pockets of high endemicity in the South-West, Centre, and the south-eastern and south-western parts of, respectively, the South and East regions, in agreement with our Fig 4E.

We mapped the predictive probability that microfilarial prevalence exceeds ≥35%, i.e., that onchocerciasis is at least mesoendemic (Fig 5). This would correspond to a predicted palpable nodule prevalence of at least 20%. For the 1971–2000 period, prior to the start of CDTI, both our study and that of Zouré et al. [33] indicated that, in Cameroon, there was a high probability of onchocerciasis being meso- to hyperendemic in the North, North-West, South-West, West and Littoral regions, as well as in the western part of the Centre region (Fig 5A). For the 2001–2010 period, we estimated high exceedance probabilities for the western part of Far North, most of Adamaoua and Centre, the southeastern part of the South and the western part of the East regions (Fig 5B). This is in agreement with the environmental suitability mapping of Cromwell et al. [48], with the exception of the southernmost part of the East region, for

which we predict moderate to high exceedance probability whilst this area appears as of low suitability for onchocerciasis transmission and is predicted to be non-endemic according to other MBG studies [33,34]. For the 2011–2020 period, our exceedance probabilities were highest in pockets of the Far North, Centre and Southwest regions. Although Schmidt et al. [34] do not present exceedance probabilities, these authors state that mean estimates of micro-filarial prevalence in Cameroon for 2018 exceeded 5% at the national level and 25% in focal areas, including in the Centre and Southwest regions [34].

## Implications of our findings for onchocerciasis elimination in Cameroon

Our results identify areas of predicted persistent transmission in the South-West, Littoral, and Centre regions for 2011–2020. Studies conducted in the South-West region had reported high levels of systematic non-adherence (16% [49] and 18% among adults [50]), owing to fears and rumours of side-effects and fatalities in areas of loiasis co-endemicity, although the prevalence and intensity of *L. loa* infection was relatively low (recorded history of eye-worm passage of 10–19% [22,51]). After 10–12 years of treatment, *O. volvulus* microfilarial prevalences above 40% were recorded in 23/29 (79%) study communities [52], in agreement with our results for the 2001–2010 decade (Fig 4C). In the West region, ongoing transmission was recorded after 15 years of treatment [53]. The proportion of systematic non-adherers was 7.4% despite a low risk of SAEs due to loiasis co-endemicity [54]. An audit of CDTI coverage and adherence in the West, Littoral and Centre regions revealed that, among those aged ≥10 years, about 10% had never taken ivermectin [55]. These studies highlight, on the one hand, the influence of community perceptions about treatment in loiasis co-endemic areas and, on the other, that not only is it important to achieve and sustain high levels of therapeutic coverage, but also to minimize the fraction of the population that never receives and takes treatment. Sub-groups of the population that remain untreated can act as reservoirs of infection and detrimentally impact progress towards elimination [56].

In addition to programmatic factors, transmission intensity also plays a crucial role in the persistence of infection hotspots. In Bafia (Centre region), annual biting rate (ABR) values ranging from 8,000 to 184,000 bites/person/year, and annual transmission potential (ATP) values of up to 13,500 L3/person/year had been recorded in 1993–1994, prior to CDTI [57]. More recent studies in the same areas measured ABRs of 233,167 – 606,370 with ATPs of 2,360 – 4,488 following 16 years of CDTI [58], and ABRs of 58,000 – 274,200 with ATPs of 44 – 255 after 20 years [59]. Another study in the same region reported daily black-fly densities of 1,113 flies/person/day, and monthly transmission potential (MTP) values of 340 – 471 L3 (head)/person/month following two decades of treatment [60]. Communities in this area had had a microfilarial prevalence greater than 80% in 1991–1993 [61], indicating holoendemicity at baseline. In the South-West region, where pre-control microfilarial prevalence had ranged from 60% (hyperendemicity) to more than 90% (holoendemicity), daily biting rates of 200 – 1,624 bites/person/day, and MTPs of 90 – 1,212 L3/person/month were reported after 10 years of CDTI [52]. A recent systematic review and meta-analysis of published studies reporting on onchocerciasis status across SSA found that holoendemicity at baseline was statistically significantly associated with increased risk of ongoing trans-mission, with some ongoing transmission foci having received more than two decades of ivermectin treatment [29].

Our exceedance map for 2011–2020 highlights areas in the western part of the Centre region, and the northern and eastern part of the South-West region, with a high predicted probability that microfilarial prevalence remained above 35% (Fig 5C). In such transmission hotspots, alternative and complementary interventions will likely be necessary, such as treat-ment with moxidectin [62] and vector control [63].

## Limitations

We relied on survey data that were not originally collected with the purpose of mapping the distribution of onchocerciasis prevalence across the country and its temporal trends after the start of interventions. Surveys were often conducted in areas where the infection was deemed to be present and/or most prevalent, making it difficult to obtain a random distribution of survey data, or to collate ground-truth data in overlooked, low-prevalence or non-endemic areas. Most data consisted of cross-sectional surveys conducted at a particular time, with surveys rarely repeated periodically in the same locales, hampering the spatio-temporal modelling of infection trends [64]. Giorgi et al. [65] have proposed a generalised linear geostatistical modelling framework to overcome this issue and exploit both the temporal and spatial correlation structure of combined data.

In addition, different areas may have started interventions on slightly different years as CDTI was rolled out in the country (interventions began in Cameroon at the end of the 1990s). Therefore, although we visualized the data following the introduction of CDTI in four 5-year periods (Fig 2), for the purposes of the MBG analysis we grouped the data into three categories, baseline (1971–2000) and two subsequent decades (2001–2010, and 2011–2020), analysing these periods separately (Tables B–D in S1 File). How well the prevalence surveys captured transmission for these models is strongly dependent on how stable transmission was during the study periods.

We did not include socio-economic covariates in our analysis because onchocerciasis distribution and prevalence tend to be highly focal and mostly limited to rural areas proximal to fast-flowing rivers and streams suitable to vector breeding, indicating that environmental covariates take pre-eminence [34,48]. In this regard, a salient limitation of our work is that we did not include distance to rivers as a covariate, which has been shown to be statistically significantly and negatively associated with microfilarial prevalence [28,34,66]. A study conducted in the Vina valley of the North region had shown that both ABR and proportion of parous flies decreased with distance from breeding sites [67], and another study conducted in the Mbam valley of the Centre region reported that the community microfilarial load also decreased with increasing distance from the river [68]. A vector density gradient was confirmed in the Mbam river, with the highest ABR and ATP values being recorded at the riverside [60]. Further refinements of our MBG approach should include this variable [33].

The paucity of survey data for some periods (2006–2010, 2016–2020) increased the uncertainty surrounding the predicted prevalence distribution in some areas, particularly in the Far North and East regions (Fig 4B, 4D, and 4F). Therefore, for these areas, our results rely solely on the value of the intercept and the effects of the covariates observed in other areas with similar environmental characteristics. However, this data paucity may have also been partly due to low environmental suitability in the northernmost and eastern part of Far North and in the southernmost and eastern part of the East region [32,48]. In consequence, our relatively high 2011–2020 predicted prevalence and exceedance probability for these areas (Figs 4E and 5C) need to be interpreted with caution (see [33,34,48] for a comparison). Environmental suitability should be considered in future improvements of our geostatistical analysis. Another factor that may have contributed to the scarcity of data for 2016–2020 is the publication in 2016 of the WHO guidelines for stopping MDA and verifying elimination of onchocerciasis [69]. These guidelines advocate the use of serology for surveillance (in children samples) in view of increased reluctance of populations in endemic communities to be skin-snipped for detection and enumeration of microfilariae, and decreased sensitivity of skin-snip microscopy as infection prevalence and intensity decline under CDTI [69]. Given that our data comprised microfilarial or nodule prevalence surveys, the dwindling sensitivity of these diagnostics under long-term CDTI is an important limitation [70,71].

We have not attempted to link our results with population densities and rates of population growth for each period in order to estimate numbers of people at risk and infected in each region over time, as well as mean prevalence values across the country for the study periods, as done by others [28,33,34,66]. This will be crucial to enable forecasting of treatment needs, number of ivermectin doses (or doses for any other alternative or complementary drug that may be adopted), and for conducting economic evaluations.

*Future research avenues* Our ultimate goal is to couple our spatiotemporal MBG analysis with transmission dynamics modelling to project times to elimination with current and alternative and/or complementary interventions. Preliminary (unpublished) results using the stochastic, individual-based EPIONCHO-IBM model [72], informed by baseline geostatistical maps fitted to estimate transmission parameters, suggest that adopting a biannual (twice-yearly) ivermectin strategy with increased coverage (80% of total population) from 2026 would lead to reaching a microfilarial prevalence <1% in 10.5 (9–12) years in Far North, North and Adamaoua regions, whilst switching to biannual moxidectin at the same coverage would reduce this time by 20%–40% (depending on the assumed cumulative sterilising effect of the drugs), or make it possible in areas that otherwise would not reach <1% microfilarial prevalence (e.g., some areas of North-West, South-West, West and Littoral) (Mathew Dixon-Zegeye and Aditya Ramani, pers. comm.). Modelling of anti-vectorial interventions [73,74] to estimate efficacy, frequency and duration necessary to accelerate progress towards EOT could provide crucial information to guide intervention efforts in areas of the Centre and South-West regions where vector biting rates have remained very high [58–60].

Entomological studies such as those undertaken in the Centre and South-West regions [52,57–60] are scarcer for other regions and should be undertaken. The importance of conducting such studies is two-fold: not only do they provide key data on vector density, infection and infectivity for entomological monitoring of the programme, but also they help to understand secular trends in vector abundance, which are crucial for transmission models. In the North and Adamaoua regions, declining trends in blackfly densities have been observed in Vina-du-Nord and Vina-du-Sud, respectively (Alfons Renz, pers. comm.) which, together with an absence of loiasis [22,51], may have contributed to better control and elimination prospects in these regions (Figs 4E and 5C). There had been a general decrease in biting rates in the Vina valley of the North region due to increasing drought and reduced water-flow in the main rivers, especially during the dry season [67]. In other foci of SSA, vector elimination due to anti-vectorial measures, and vector disappearance or long-term declines in vector density due to ecological change have contributed to elimination of transmission [29].

In areas of lower onchocerciasis (hypo- to meso-) endemicity that co-occur with high loiasis endemicity [22,23,51], a test-and-not-treat strategy (test for *L. loa* microfilaraemia levels and do not treat with ivermectin if these exceed 20,000 microfilariae per millilitre of blood) has been successfully piloted in the Okola health district of the Centre region [75]. Individuals who are excluded from ivermectin treatment can be offered doxycycline (which at a daily dose of 100 mg for 5–6 weeks is macrofilaricidal for *O. volvulus* [76,77] but does not affect *L. loa* (which lacks *Wolbachia* endosymbiotic bacteria) [78], circumventing the SAE problem [24,25,76]. For areas of the lower half of the country which are co-endemic with highly-endemic loiasis, further modelling of test-and-not-treat [79] for a number of years, of a subsequent switch to ivermectin or moxidectin MDA when/where possible [62], and of treatment with doxycycline [80] would help inform the best strategies to accelerate EOT.

We also recommend that additional surveys be conducted in areas of greater uncertainty, potentially using adaptive sampling, informed by the results presented here. Adaptive sampling has been discussed both in the context of a sequence of prevalence surveys to assess the progress of an MDA programme (where survey designs appropriate to the early stages can be

adapted in response to spatially-heterogeneous changes in prevalence over consecutive treatment rounds), and in post-treatment and post-elimination surveillance [81].

## Conclusions

Our MBG analysis of the microfilarial prevalence of *O. volvulus* from 1971 to 2020 is broadly consistent with other MBG studies and indicates that Cameroon is making progress in its quest to eliminate onchocerciasis transmission, but areas of stubborn persistence remain which will likely require alternative and/or complementary interventions. We anticipate that the combination of geostatistical and transmission dynamics modelling will assist the Cameroon National Committee for the Elimination of Onchocerciasis in their recommendations to MINSANTE and the NOCP to reach elimination targets.

## Supporting information

**S1 File. Detailed methods. Table A. Definition of the bioclimatic variables used. Text A. Conversion of nodule prevalence into microfilarial prevalence. Text B. Description of the Model-Based Geostatistical (MBG) approach used and parameter estimation procedures. Table B. Parameter estimates and 95% confidence limits (CL) for the binomial geostatistical model of the association between microfilarial prevalence and environmental covariates for the baseline (1971–2000) period, and spatial covariance parameters. Table C. Parameter estimates and 95% confidence limits (CL) for the binomial geostatistical model of the association between microfilarial prevalence and environmental covariates for the 2001–2010 period, and spatial covariance parameters. Table D. Parameter estimates and 95% confidence limits (CL) for the binomial geostatistical model of the association between microfilarial prevalence and environmental covariates for the 2011–2020 period, and spatial covariance parameters. Text C. Interpretation of environmental coefficients.**
(PDF)

## Acknowledgements

We would like to thank the Cameroon Ministry of Public Health and the National Onchocerciasis Control Programme, who provided a wealth of information and data that enabled us to understand the evolution of control activities and onchocerciasis endemicity trends in Cameroon. We would also like to thank Dr Alfons Renz (University of Tübingen, Germany) for insights into the declining biting rates in the North and Adamoua regions, and Dr Mathew Dixon-Zegeye (Imperial College London, UK) and Aditya Ramani (Royal Veterinary College, UK) for valuable discussions on EPIONCHO-IBM-projected microfilarial trends for Cameroon.

## Author contributions

**Conceptualization:** Yannick Niamsi-Emalio, Hugues C. Nana-Djeunga, Claudio Fronterrè, Joseph Kamgno, María-Gloria Basáñez.

**Data curation:** Yannick Niamsi-Emalio, Hugues C. Nana-Djeunga.

**Formal analysis:** Yannick Niamsi-Emalio, Claudio Fronterrè, Himal Shrestha, María-Gloria Basáñez.

**Funding acquisition:** Joseph Kamgno, María-Gloria Basáñez.

**Investigation:** Yannick Niamsi-Emalio, Hugues C. Nana-Djeunga.

**Methodology:** Yannick Niamsi-Emalio, Hugues C. Nana-Djeunga, Claudio Fronterrè, María-Gloria Basáñez.

**Project administration:** Joseph Kamgno, María-Gloria Basáñez.

**Resources:** Hugues C. Nana-Djeunga, Georges B. Nko'Ayissi, Théophile M. Mpaba Minkat, Joseph Kamgno, María-Gloria Basáñez.

**Supervision:** Hugues C. Nana-Djeunga, Claudio Fronterrè, Joseph Kamgno, María-Gloria Basáñez.

**Visualization:** Yannick Niamsi-Emalio, Himal Shrestha, María-Gloria Basáñez.

**Writing – original draft:** Yannick Niamsi-Emalio, Hugues C. Nana-Djeunga, María-Gloria Basáñez.

**Writing – review & editing:** Yannick Niamsi-Emalio, Hugues C. Nana-Djeunga, Claudio Fronterrè, Himal Shrestha, Georges B. Nko'Ayissi, Théophile M. Mpaba Minkat, Joseph Kamgno, María-Gloria Basáñez.

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
