## [Decision Letter · Decision Letter 0]

26 Oct 2024

PNTD-D-24-00734Model-Based Geostatistical Mapping of the Prevalence of Onchocerca volvulus in Cameroon between 1971 and 2020PLOS Neglected Tropical Diseases Dear Dr. Basáñez, Thank you for submitting your manuscript to PLOS Neglected Tropical Diseases. After careful consideration, we feel that it has merit but does not fully meet PLOS Neglected Tropical Diseases's publication criteria as it currently stands. Therefore, we invite you to submit a revised version of the manuscript that addresses the points raised during the review process. Please submit your revised manuscript within 60 days Dec 25 2024 11:59PM. If you will need more time than this to complete your revisions, please reply to this message or contact the journal office at plosntds@plos.org. Please include the following items when submitting your revised manuscript:* A rebuttal letter that responds to each point raised by the editor and reviewer(s). You should upload this letter as a separate file labeled 'Response to Reviewers '. This file does not need to include responses to any formatting updates and technical items listed in the 'Journal Requirements' section below.* A marked-up copy of your manuscript that highlights changes made to the original version. You should upload this as a separate file labeled 'Revised Manuscript with Track Changes '.* An unmarked version of your revised paper without tracked changes. You should upload this as a separate file labeled 'Manuscript '. If you would like to make changes to your financial disclosure, competing interests statement, or data availability statement, please make these updates within the submission form at the time of resubmission. Guidelines for resubmitting your figure files are available below the reviewer comments at the end of this letter. We look forward to receiving your revised manuscript. Kind regards, Francesca TamarozziSection EditorPLOS Neglected Tropical Diseases Francesca TamarozziSection EditorPLOS Neglected Tropical Diseases

Shaden Kamhawi

co-Editor-in-Chief

Paul Brindley

co-Editor-in-Chief

 **Journal Requirements:** **Additional Editor Comments (if provided):** 

**Reviewers' Comments:** Reviewer's Responses to Questions

**Key Review Criteria Required for Acceptance?**

**Methods**

-Are the objectives of the study clearly articulated with a clear testable hypothesis stated?

-Is the study design appropriate to address the stated objectives?

-Is the population clearly described and appropriate for the hypothesis being tested?

-Is the sample size sufficient to ensure adequate power to address the hypothesis being tested?

-Were correct statistical analysis used to support conclusions?

-Are there concerns about ethical or regulatory requirements being met?

Reviewer #1: (No Response)

Reviewer #2: Clear study objectives and time periods. The overview of the study population, unit of analysis and data sources used for modelling which could be improved (a couple of minor comments below). Suitable MBG approach followed for the analysis though a few points could be explained more clearly about the model selection. No ethical concerns.

Specific comments:

-Data overview:

oI appreciate this analysis is being conducted on data from a diverse range of different studies already described in the systematic review, but it would be helpful to be given a brief overview of what kind of studies these consist of i.e. are they all school/household/community prevalence surveys, what diagnostics were used etc. To help the reader understand the complexities of jointly modelling all of this data together, particularly if anything changes over the study periods in terms of how data was collected that could have biased the results.

oI would include the spatial resolution of the environmental covariates used in the analysis.

-Would be helpful to include in the article or supplementary material how nodule prevalence was converted into microfilarial prevalence, not just cited (lines 190-191).

-‘the microfilarial prevalence endemicity levels described in [27] were used for stratification.’ (lines 196-197), please state the endemicity levels used in the analyses here.

-Lines 200-201: ‘We identified a total of 571 georeferenced prevalence survey data-points for the 1971–2000 period; 410 for 2001– 2005; 98 for 2006–2010; 261 for 2011–2015 and 64 for 2016–2020, for a total of 1,404.’ – This isn’t clear, are you saying that there were 1,404 prevalence survey datapoints geolocated at 571 unique locations? It’s also not clear what a datapoint is here – is it an individual, a school? Please explain to the reader to give an idea of the unit of analysis and sample size.

-Line 203 – I was unclear on the spatial unit of analysis here. You describe having 571 locations but then say the unit is health districts in Cameroon. It would be helpful for the readers not familiar with Cameroon to be given an idea of the size of a health district to understand the spatial resolution of your analyses/limitations of assumed location. Seems from what I’ve found online that the country is broken up into 189 health districts which doesn’t fit with your locations. Please clarify and explicitly explain any assumptions/approximations/aggregations for the location used in the analysis.

-Covariate selection (lines 217-219) is not clearly described. I would suggest changing the order of this section and explain how variables were selected with GLMs before moving to fitting the full binomial geostatistical model (I would also suggest specifying that this is the model that you fit rather than just the PrevMap function). I also saw the use of the term ‘spatial GLM’ in your results and wanted to clarify that here you are just talking about a GLM for model selection rather than a geostatistical model?

-It also wasn’t clear to me how you selected variables – did you fit a single GLM with all the variables in it and then just select those that had p<0.05? A better approach would be assessing goodness of fit e.g. with AIC or cross-validation.

-I would also move ‘The degree of significance for each variable was set at 5%’ (lines 220-221) before the multicollinearity sentence so it's clear what you define as being ‘statistically significant’.

-It would also be helpful to know how this process took place considering your data was subdivided into different periods i.e. whether you conducted covariate selection using all the data together or to define different models for the different periods. Seems from results like you fit independent geostatistical models for each period and then one model to all of the data with period as a covariate, but it would also be helpful to explain this process clearly to the reader.

-Lines 219-220: Please give the details of your multicollinearity checks i.e. what VIF cut-off did you use.

-Nugget effect – it seems like this wasn’t included in your models, but there is evidence of a nugget in your variogram C.

-Presumably you assumed an exponential correlation function, but would be good to specify this in the main text.

-Prediction – no mention of this in your methods. Presumably these were marginal predictions?

**Results**

-Does the analysis presented match the analysis plan?

-Are the results clearly and completely presented?

-Are the figures (Tables, Images) of sufficient quality for clarity?

Reviewer #1: (No Response)

Reviewer #2: Results has all the expected results in it but some tidying up of figures and text is required to make this section clearer and more specific.

Specific comments:

-Figures 1 and 2:

oWould be nice to have a scale bar on your map.

oThe prevalence scales aren’t very clear - the use of ‘00’ and the way intervals are shown.

oThe use of different prevalence scales used between the two figures also makes it challenging to compare the changes over time. I would use the same scale across both figures.

oMaking your datapoints slightly transparent would help be visualising overlap of datapoints.

oFigure 2 – The size makes this challenging to see the points clearly and I would remove the region names as they’re too small to read.

-Use of terminology:

o ‘spatial GLM’ (Line 273) – it would be clearer to call this a binomial geostatistical model. I see the term GLM is also used in line 285 too, but the supplementary tables seem to geostatistical models.

oTable 1: It would be helpful to be more specific with your terminology here e.g. give it a title ‘Parameter estimates for the binomial geostatistical model for microfilarial prevalence for the complete period 1971-2020….’

-Table 1:

oPlease include confidence intervals, more interpretable than SE and z-values. Please also include for phi and sigma^2.

oI’m not really clear on the significance of this ‘all period’ model – it doesn’t get used in the predictions, as the analysis seems to be really conducted using period-specific models. This all needs to be much clearer in the methods section.

-Figure 4 and text lines 299-305:

oPlease add a legend key to show prevalence and SE

oCaption of Figure 4 needs to be more specific e.g. Mean predicted prevalence from the period-specific binomial geostatistical models.. Please also make this clearer in the text.

oDifferent colours are used for prevalence levels for A, C and E. These need to be the same to allow for comparison between the figures.

-Line 312: This is mean predicted microfilarial prevalence. Please be careful with the terminology used.

-Exceedance probability:

oGreat to see these.

oLine 324 – typo for ‘predictive’.

oFigure 5 – check your prevalence scales, they have different colours in panel C which makes it hard to compare across them.

**Conclusions**

-Are the conclusions supported by the data presented?

-Are the limitations of analysis clearly described?

-Do the authors discuss how these data can be helpful to advance our understanding of the topic under study?

-Is public health relevance addressed?

Reviewer #1: (No Response)

Reviewer #2: I found this a long and quite challenging section to read/digest and would suggest some substantial cutting and restructuring of the discussion section to more clearly communicate the key points that the author wishes to make. There is a lot of information but it’s often delivered in quite an unstructured way without linking it to previous sentences or reflecting on it. I suggest starting each paragraph with a sentence that articulates the main point you wish to make, followed by sentences describing the evidence, giving references to literature, reflections etc.

Generally, I don’t feel that the first half of discussion did justice to the findings in the results. Much of it before line 400 reads like a results section and is hard to follow, but it is much better after this point (with lots of interesting discussion points). From the very beginning of the discussion I would expect a much clearer discussion of i) spatio-temporal variation in predicted prevalence, comparison with other study findings and considerations of changes in the disease’s epidemiology and data collection needs; ii) exceedance probability and implications for disease control; iii) considerations of data/model fit/other limitations. I suggest cutting the discussion down significantly (particularly the first half, much of which I think isn’t needed) and restructuring paragraphs so the reader can clearly identify the main takeaway points.

Specific comments:

-Paragraph 1:

oThis reads a bit too much like the methods. I think helpful to give a quick overview of your analysis, but I would suggest 2 sentences explaining the study aims/data followed by a very top-level overview of the main results from the results.

oLines 338 – ‘predicted prevalence’ not just ‘prevalence’

oLine 339 – ‘inedited data’ - typo?

oLine 340 - ‘to be presented elsewhere’ – unclear what this means?

oLine 342 - Unclear what the relevance of the comparison to reference 34 is.

-Paragraph 2:

oGiven this study was a prediction study rather than a causal inference study I would leave any interpretation of coefficients for later on in the discussion (or preferably not at all).

oI would make your second paragraph the one that discusses your most important results which are your findings about the spatial distribution of predicted prevalence, how that changes over time.

-Paragraph 3:

oIt was not clear to me what the point of this paragraph was, there didn’t seem to be any actual comparison but rather just a description of what other modelling has been done.

-Paragraph 4:

oThis wasn’t very clear to me. Again, a first sentence that explains the main point of the paragraph would be helpful and would help you to focus the paragraph more clearly. E.g. ‘Our prevalence predictions were similar to findings of a recent study..’

oCheck your referencing style for including in text citations of papers – I would expect these to include author names rather than just ‘in [33] and [34]’.

oFinal sentence of the paragraph is not linked to the rest of the text at all.

-Limitations:

oThis was better. Remember that how well your prevalence surveys captured transmission for your models is dependent on how stable transmission was during the study periods.

oLine 403-404 -how is the MBG framework designed to overcome the limitations of data not collected for mapping and temporal trends?

oLines 417-425: why wasn’t this variable included?

**Editorial and Data Presentation Modifications?**

Reviewer #1: (No Response)

Reviewer #2: (No Response)

**Summary and General Comments**

Reviewer #1: (No Response)

Reviewer #2: This is an interesting analysis of collated onchocerciasis prevalence data and much of it is well written and clear. Areas of improvement are the following: methods (clarification needed on the modelling approach and more information needed about the data used); results (some minor edits to figures and captions required to make them more interpretable; more specific language required relating to models used and mean predicted prevalence shown in plots); discussion (requires the most work – needed to cut and focus the existing text to make it more readable and to clearly communicate your key points; there are some really interesting points in this section however!).

As a result of the significant amount of editing required in the discussion section, I have recommended a major revision.

PLOS authors have the option to publish the peer review history of their article (what does this mean? ). If published, this will include your full peer review and any attached files.

**Do you want your identity to be public for this peer review?** For information about this choice, including consent withdrawal, please see our Privacy Policy .

Reviewer #1: No

Reviewer #2: No

---

## [Decision Letter · Decision Letter 1]

4 Mar 2025

Dear Professor Basáñez,

We are pleased to inform you that your manuscript 'Model-Based Geostatistical Mapping of the Prevalence of Onchocerca volvulus in Cameroon between 1971 and 2020' has been provisionally accepted for publication in PLOS Neglected Tropical Diseases.

Best regards,

Francesca Tamarozzi

Section Editor

Francesca Tamarozzi

Section Editor

Shaden Kamhawi

co-Editor-in-Chief

Paul Brindley

co-Editor-in-Chief

Reviewer's Responses to Questions

**Key Review Criteria Required for Acceptance?**

**Methods**

-Are the objectives of the study clearly articulated with a clear testable hypothesis stated?

-Is the study design appropriate to address the stated objectives?

-Is the population clearly described and appropriate for the hypothesis being tested?

-Is the sample size sufficient to ensure adequate power to address the hypothesis being tested?

-Were correct statistical analysis used to support conclusions?

-Are there concerns about ethical or regulatory requirements being met?

Reviewer #1: (No Response)

Reviewer #2: (No Response)

**Results**

-Does the analysis presented match the analysis plan?

-Are the results clearly and completely presented?

-Are the figures (Tables, Images) of sufficient quality for clarity?

Reviewer #1: (No Response)

Reviewer #2: (No Response)

**Conclusions**

-Are the conclusions supported by the data presented?

-Are the limitations of analysis clearly described?

-Do the authors discuss how these data can be helpful to advance our understanding of the topic under study?

-Is public health relevance addressed?

Reviewer #1: (No Response)

Reviewer #2: (No Response)

**Editorial and Data Presentation Modifications?**

Reviewer #1: (No Response)

Reviewer #2: (No Response)

**Summary and General Comments**

Reviewer #1: (No Response)

Reviewer #2: The authors have carefully responded to all of my queries and I am satisfied that they have addressed all of the issues identified. The manuscript is clearer and congratulations to the authors on their hard work!

PLOS authors have the option to publish the peer review history of their article (what does this mean? ). If published, this will include your full peer review and any attached files.

**Do you want your identity to be public for this peer review?** For information about this choice, including consent withdrawal, please see our Privacy Policy .

Reviewer #1: No

Reviewer #2: No

---

## [Editor Report · Acceptance letter]

Dear Professor Basáñez,

We are delighted to inform you that your manuscript, "Model-Based Geostatistical Mapping of the Prevalence of Onchocerca volvulus in Cameroon between 1971 and 2020," has been formally accepted for publication in PLOS Neglected Tropical Diseases.

Best regards,

Shaden Kamhawi

co-Editor-in-Chief

Paul Brindley

co-Editor-in-Chief
